# Oxidative Ferritin Destruction: A Key Mechanism of Iron Overload in Acetaminophen-Induced Hepatocyte Ferroptosis

**DOI:** 10.3390/ijms26157585

**Published:** 2025-08-05

**Authors:** Kaishuo Gong, Kaiying Liang, Hui Li, Hongjun Luo, Yingtong Chen, Ke Yin, Zhixin Liu, Wenhong Luo, Zhexuan Lin

**Affiliations:** 1Bio-Analytical Laboratory, Shantou University Medical College, No. 22, Xinling Road, Shantou 515041, China; 21ksgong@alumni.stu.edu.cn (K.G.); 19kyliang@stu.edu.cn (K.L.); hli@stu.edu.cn (H.L.); hjluosumc@stu.edu.cn (H.L.); 22ytchen@stu.edu.cn (Y.C.); 24kyin@stu.edu.cn (K.Y.); 20zxliu3@stu.edu.cn (Z.L.); 2Shantou Key Laboratory for New Drugs Research & Development, No. 22, Xinling Road, Shantou 515041, China

**Keywords:** acetaminophen, hepatotoxicity, ferroptosis, ferritin destruction, iron overload, oxidative stress

## Abstract

Although acetaminophen (APAP) overdose represents the predominant cause of drug-induced acute liver failure (ALF) worldwide and has been extensively studied, the modes of cell death remain debatable and the treatment approach for APAP-induced acute liver failure is still limited. This study investigated the mechanisms of APAP hepatotoxicity in primary mouse hepatocytes (PMHs) by using integrated methods (MTT assay, HPLC analysis for glutathione (GSH), Calcein-AM for labile iron pool detection, confocal microscopy for lipid peroxidation and mitochondrial superoxide measurements, electron microscopy observation, and Western blot analysis for ferritin), focusing on the role of iron dysregulation under oxidative stress. Our results showed that 20 mM APAP treatment induced characteristic features of ferroptosis, including GSH depletion, mitochondrial dysfunction, and iron-dependent lipid peroxidation. Further results showed significant ferritin degradation and subsequent iron releasing. Iron chelator deferoxamine (DFO) and N-acetylcysteine (NAC) could alleviate APAP-induced hepatotoxicity, while autophagy inhibitors did not provide a protective effect. In vitro experiments confirmed that hydrogen peroxide directly damaged ferritin structure, leading to iron releasing, which may aggravate iron-dependent lipid peroxidation. These findings provide evidence that APAP hepatotoxicity involves a self-amplifying cycle of oxidative stress and iron-mediated oxidative damaging, with ferritin destruction playing a key role as a free iron source. This study offers new insights into APAP-induced liver injury beyond conventional cell death classifications, and highlights iron chelation as a potential therapeutic strategy alongside traditional antioxidative treatment with NAC.

## 1. Introduction

Acetaminophen (APAP), also known as paracetamol, is one of the most widely used over-the-counter (OTC) antipyretic and analgesic drugs. At therapeutic doses (up to 4 g/day for healthy adults), APAP is considered safe and effective. However, its wide availability and unsupervised use contribute to a high risk of overdose, leading to excessive hepatic formation of N-acetyl-p-benzoquinone imine (NAPQI)—the reactive metabolite that initiates acute liver failure (ALF) characterized by centrilobular necrosis [1]. APAP-induced hepatotoxicity remains the most common cause of ALF in developed countries, accounting for over 50% of ALF cases in the United States and Europe [2,3]. APAP abuses often happen in home-resident patients during a pandemic, for example, APAP consumption dramatically increased on a worldwide scale after the emergence of the COVID-19 pandemic, thus elevating the risk of liver injury [4]. Although APAP is the most extensively studied hepatotoxic drug worldwide, the treatment options for APAP-induced ALF remain limited. N-acetyl cysteine (NAC) is a thiol-containing compound and the only therapeutic option for APAP-overdosed patients approved by the United States Food and Drug Administration. Its hepatoprotective mechanisms involve serving as a cysteine precursor to restore hepatic GSH pools for NAPQI detoxification, as a free radical scavenger to mitigate oxidative stress, and converting to Krebs cycle intermediates that enhance mitochondrial ATP production in hepatocytes [5]. However, this medication has adverse effects and a relatively narrow therapeutic window [6]. Without timely intervention, advanced liver injury may occur, and liver transplantation often becomes the only viable option for these patients [7,8,9].

The current understanding of APAP hepatotoxicity centers on the formation of a reactive metabolite, N-acetyl-p-benzoquinone imine (NAPQI), primarily generated via the hepatic cytochrome P450 2E1 (CYP2E1) metabolism. This oxidative pathway represents only a minor fraction (5–9%) of total APAP clearance at therapeutic doses, where 85–90% undergoes detoxification through uridine diphosphate glucuronosyltransferase-mediated glucuronidation or sulfotransferase-catalyzed sulfation to renally excreted conjugates [10]. Under normal conditions, NAPQI is rapidly detoxified by conjugation with glutathione (GSH) [11,12,13,14,15,16]. However, during APAP overdose, saturation of glucuronidation and sulfation pathways diverts substantially more substrate toward phase I metabolism, dramatically amplifying NAPQI generation [10]. This metabolic shift leads to excessive NAPQI accumulation, which would then cause a depletion of GSH or form protein adducts (particularly on mitochondrial proteins), triggering oxidative damage and subsequent cell death [8,17,18,19]. In spite of evidences showing existence of a general oxidative stress during APAP hepatotoxicity, the modes of cell death remain debatable [20], which leads to controversial conclusions and ultimately hamper the translation of new therapeutic approaches.

The mode of cell death in APAP hepatotoxicity was previously termed oncotic necrosis, due to the extensive cell swelling and membrane rupture [21]. However, studies revealed features of apoptosis, including mitochondrial release of apoptosis-inducing factor (AIF) and endonuclease G, and DNA fragmentation [22,23]. Additionally, autophagy has been implicated in APAP hepatotoxicity [24,25,26]. It can remove damaged mitochondria and APAP-adducted proteins, thereby mitigating injury [24,27]. While, in vitro experiment in fibroblasts and cancer cells showed that excessive autophagy might promote erastin-induced ferroptosis by degrading ferritin and increasing intracellular free iron levels [26]. But there is no literature about autophagy-induced degradation of ferritin in APAP-induced hepatoxicity. Evidently, no single cell death modes (necroptosis, ferroptosis, apoptosis, or autophagic cell death) fully explain APAP hepatotoxicity. Therefore, elucidating the critical mechanisms in APAP hepatotoxicity rather than merely defining modes of cell death has more realistic significance for discovering novel therapeutic targets and improving clinical outcomes.

Notably, GSH depletion and the subsequent oxidative stress have been considered to be the initial events in APAP hepatotoxicity [6,14], with accumulating evidence implicating elevated intracellular free iron levels in APAP hepatotoxicity [24,27]. Transition metals like iron and copper could catalyze Fenton reactions [28,29], promote GSH oxidation [30] and generate highly reactive hydroxyl radicals, leading to aggravating oxidative stress and lipid peroxidation [31]. These metal-mediated processes have been shown to trigger multiple regulated cell death pathways, including ferroptosis, cuproptosis, apoptosis, autophagy, necroptosis and pyroptosis [32]. Therefore, investigating the origin of excessive free iron in hepatocytes following APAP exposure may be a crucial research direction for further understanding APAP-induced liver injury. Consequently, this study aimed to investigate the mechanisms of iron dysregulation and its contribution to cell death in APAP-induced hepatotoxicity using primary mouse hepatocytes (PMHs). Specifically, we sought to determine (1) whether ferroptosis is a significant pathway in APAP-treated PMHs; (2) the role of ferritin degradation as a source of pathogenic free iron; (3) the potential link between APAP-induced oxidative stress and ferritin damage; and (4) the therapeutic relevance of iron chelation alongside antioxidant treatment. By elucidating this self-amplifying cycle of oxidative stress and iron-mediated damage, we provide new mechanistic insights beyond conventional cell death classifications and highlight combined iron chelation and antioxidative therapy as a potential strategy for APAP overdose.

## 2. Results

### 2.1. The Hepatotoxicity Induced by APAP

The results of MTT assay showed that the cell viability of PMHs was 101.1 ± 1.5% in the control group, and 100.1 ± 1.9%, 100.8 ± 1.9%, 99.7 ± 1.0%, 100.2 ± 1.9%, and 99.0 ± 1.3% the in 10–30 mM APAP treatment groups, respectively, at the 2 h time point, showing no significant differences among various groups. After treatment with 10–30 mM APAP, the cell viabilities of PMHs decreased significantly for a longer duration of time (4 h, 6 h, 8 h, 10 h, and 12 h), compared with the corresponding control (*p* < 0.05) (Figure 1A). Based on these results, APAP with a concentration of 20 mM was selected and used in the following experiments. DFO (0.25–2 mM, iron chelator) or NAC (1–4 mM, antioxidant) could, in a concentration-dependent manner, alleviate the toxic effect of 20 mM APAP (*p* < 0.05) (Figure 1B,C). In contrast, chloroquine (25–100 μM, autophagy inhibitor) or leupeptin (50–300 μM, protease inhibitor) treatment did not significantly inhibit the toxic effect of 20 mM APAP (*p* > 0.05) (Figure 1D,E).

### 2.2. The Changes in PMHs’ Ultrastructure After APAP Treatment

Transmission electron microscopic images of control PMHs showed intact mitochondria with bean-shaped structures and numerous transversely orientated cristae enveloped by an intact outer membrane (Figure 2A). After treatment with 20 mM APAP, the PMHs exhibited spherical mitochondria with mitochondrial membrane disarrayed cristae and a decreased number of mitochondria and electron density of the matrix (Figure 2B,E). Loss of cell nuclear membrane integrity and disintegration of organelles were also observed (Figure 2B). DFO (2 mM) or NAC (2 mM) treatment could restore cell nuclear membrane integrity and mitochondrial structures to some extent, and the number of mitochondria as well (Figure 2C–E). No apoptotic bodies were observed in APAP-treated PMHs.

### 2.3. Decrease in Intracellular GSH and Increase in LIP Caused by Treatment with APAP

APAP (20 mM) treatment significantly depleted the cellular GSH. NAC (2 mM) or DFO (2 mM) treatment could significantly restore intracellular GSH level (*p* < 0.0001) (Figure 3A). In addition, the intracellular GSH level was significantly higher in the 20 mM APAP + 2 mM NAC group than that in the 20 mM APAP + 2 mM DFO group (*p* < 0.0001). Intracellular labile iron pool (LIP) is a pool of chelatable and redox-active iron, which was determined by fluorescence probe Calcein-AM. Treatment with 20 mM APAP for 12 h caused a significant elevation of fluorescence intensity in PMHs (*p* < 0.001), with 0.88 ± 0.08 in the control group and 2.97 ± 0.12 in the APAP-treated group, respectively (Figure 3B). NAC (2 mM) or DFO (2 mM) could significantly inhibit the increase in LIP induced by 20 mM APAP (*p* < 0.0001).

### 2.4. Lipid Peroxidation and Mitochondrial Superoxide Production in PMHs After APAP Exposure

To determine the site of ROS generation, redox-sensitive dye BODIPY-C11 was used to detect lipid ROS (Figure 4A). When BODIPY-C11 is reduced, this dye exhibits red fluorescence and when oxidized, its emission peak shifts to the green spectrum. Treatment of PMHs with 20 mM APAP for 4 h increased ROS generation as measured by the increased BODIPY-C11 green fluorescence intensity (Figure 4A,B). NAC (2 mM) or DFO (2 mM) could significantly decrease green fluorescence intensity, suggesting the decrease in lipid ROS (Figure 4A,B).

Fluorescence probe MitoSOX Red specifically targeting superoxide anions, the predominant ROS generated in the mitochondria, is used to detect mitochondrial ROS (Figure 4C,D,E). At the same time, MitoTracker probes (green) were used to stain active mitochondria in PMHs. As shown in Figure 4C, control cells displayed intense green fluorescence with minimal red fluorescence, suggesting abundant mitochondria and low MitoSOX levels under physiological conditions. In contrast, hepatocytes treated with 20 mM APAP exhibited markedly enhanced red fluorescence, with particularly intense staining observed in nuclear regions. Concurrently, the green fluorescence intensity was substantially diminished with the emergence of nuclear region staining, indicative of both mitochondrial impairment and elevated superoxide production. Treatment with either 2 mM NAC or 2 mM DFO effectively attenuated the red fluorescence intensity while restoring green fluorescence intensity.

### 2.5. Effects of APAP on Hepatic Ferritin (FTH1) Expression

The results of Western blot showed that APAP treatment significantly reduced ferritin heavy chain 1 (FTH1) protein levels in PMHs, compared with the control group (*p* < 0.05, Figure 5). This APAP-induced depletion of FTH1 was effectively attenuated by co-treatment with either 2 mM NAC or 2 mM DFO, both showing statistically significant protective effects (*p* < 0.05) (Figure 5B). In contrast, neither 100 μM CRO (autophagy inhibitor) nor 200 μM leupeptin (protease inhibitor) demonstrated any significant modulation of the APAP-mediated FTH1 reduction (Figure 5C).

### 2.6. Iron Release from Ferritin Under Oxidative Stress

As demonstrated in Figure 6, the control group (0.25 mg/mL ferritin + calcein) showed intact ferritin could not significantly decrease the fluorescence intensity of calcein (Figure 6). The H_2_O_2_ control group (containing 10 μM H_2_O_2_ and calcein but no ferritin) also could not significantly decrease the fluorescence intensity of calcein. The H_2_O_2_ treatment group (0.25 mg/mL ferritin + 10 μM H_2_O_2_ + calcein) displayed rapid fluorescence quenching during the first 60–120 s, indicating iron release from ferritin by the action of H_2_O_2_.

## 3. Discussion

In the present study, primary mice hepatocytes (PMHs) were used to establish a robust APAP-induced hepatotoxicity paradigm through viability assays (MTT) and morphological evaluations (Appendix A). PMHs have been demonstrated to express CYP2E1—the principal enzyme responsible for the formation of reactive metabolite NAPQI from APAP. This well-characterized metabolic capacity establishes PMHs as a physiologically relevant model for APAP-induced hepatotoxicity research [10,33,34]. Our data demonstrated that 10–30 mM APAP induced time and concentration-dependent cytotoxicity in PMHs. Interestingly, the dose–response relationship became less pronounced at higher concentrations (20–30 mM) with prolonged exposure (>10 h). This attenuation may reflect saturation of cytochrome P450-mediated metabolic activation, as the conversion of APAP to its toxic metabolite NAPQI is ultimately limited by the enzymatic capacity of hepatic CYP isoforms, particularly CYP2E1. This phenomenon is mechanistically supported by transcriptional downregulation of key CYPs (CYP2E1, CYP1A2, and CYP3A4) in hepatocytes exposed to 20 mM APAP for >6 h [34].

Both N-acetylcysteine (NAC) and deferoxamine (DFO, an iron chelator) significantly attenuated APAP-induced cell death, whereas autophagy inhibitors (chloroquine/leupeptin) did not show protective effects. This differential response implies that oxidative stress and iron dysregulation, rather than autophagic processes, could be the primary drivers of APAP-induced cytotoxicity.

Ferroptosis is characterized by GSH depletion and iron-dependent lipid peroxidation [1,35,36]. Consistent with these studies, our results showed that APAP significantly reduced intracellular GSH, which was reversed by NAC and DFO (Figure 3A). NAC might replenish GSH by supplying cysteine (an essential precursor in glutathione production) or directly bind to toxic NAPQI and scavenge free radicals [37]. Surprisingly, DFO also restored GSH levels, suggesting iron-catalyzed oxidative consumption of GSH might be another significant GSH depletion pathway. Iron, as a transition metal element, can catalyze the generation of reactive oxygen species (ROS) through Fenton reaction intracellularly, and act as an essential cofactor in lipid peroxidation cascades [38]. We observed elevated labile iron pool (LIP) levels in APAP-treated cells, alongside increased mitochondrial superoxide (MitoSOX Red) and lipid peroxidation (C11-BODIPY581/591). Both NAC and DFO attenuated these effects, though only DFO directly chelates iron. Intriguingly, NAC also reduced LIP, suggesting it may limit free iron releasing indirectly. Our ultrastructural analysis revealed that APAP caused characteristic mitochondrial damage, including cristae disorganization (Figure 2B). These morphological changes are consistent with the observed mitochondrial superoxide overproduction (Figure 4C) and lipid peroxidation (Figure 4A,B), which are considered to be characteristic events of ferroptosis. The absence of apoptotic bodies in our ultrastructural observation supports the predominance of necrotic cell death in APAP hepatotoxicity that was defined in previous studies [39]. Both NAC and DFO preserved mitochondrial integrity (Figure 2C,D), implicating the important role of oxidative stress and iron dysregulation in mitochondrial damaging.

Since iron-mediated amplification of hepatocytes injury is important, the origin of iron overload during APAP toxicity is explored in the present work. In biological systems, cellular iron exists in two forms: (1) not chelatable iron, mainly stored in ferritin, and (2) labile iron, which includes free cytoplasmic chelatable iron, forming a dynamic labile iron pool (LIP) that actively participates in pathological and physiological processes [40,41]. Ferritin serves as the major intracellular iron storage protein, composed of a hollow spherical structure with a protein shell and an iron core [41]. The protein shell consists of 24 highly symmetrical subunits, including ferritin heavy chain (FTH) and ferritin light chain (FTL). Within its central cavity, ferritin binds varying quantities of Fe^3+^ [42]. Under physiological conditions, ferritin can sequester up to 2000 Fe^3+^ ions, reaching a maximum storage capacity of approximately 4500 ions when fully saturated [43]. If ferritin integrity is compromised pathologically, iron stored within its core could be released into the cytoplasm, expanding the LIP. Our results demonstrated that APAP treatment downregulated FTH1 expression in hepatocytes, indicating a reduction in the intracellular iron-binding capacity of ferritin. This impairment in ferritin-mediated iron sequestration likely contributes to the observed expansion of the LIP, suggesting a key mechanism for APAP-induced iron dysregulation. A previous study has reported that autophagy may contribute to ferritin degradation, thereby promoting iron release [25]. However, co-localization studies in our work revealed minimal lysosome–ferritin interaction post-APAP treatment (Appendix A), and neither autophagy inhibitors (chloroquine, CRO) nor protease inhibitors (leupeptin) restored FTH1 levels, ruling out autophagic turnover and lysosomal protease-mediated degradation of ferritin in the APAP hepatotoxicity. An in vitro experiment using the iron-sensitive fluorescent probe Calcein demonstrated that hydrogen peroxide (H_2_O_2_) directly disrupts ferritin, leading to iron release. Since Calcein fluorescence is quenched upon iron binding, the observed decrease in fluorescence intensity confirms ferritin can undergo oxidative damage by H_2_O_2_, leading to subsequent iron liberation. This finding also provides insight into why NAC, a potent antioxidant, effectively reduces LIP levels in the cell culture experiment, presumably by scavenging H_2_O_2_ and preventing ferritin from oxidative damaging. Therefore, oxidative stress-induced ferritin destruction might serve as the critical link between primary oxidative injury and secondary iron overload in APAP-induced hepatotoxicity.

## 4. Materials and Methods

### 4.1. Reagents

Reagents acetaminophen, dimethyl sulfoxide (DMSO), 4-(2-hydroxyethyl)-1-piperazineethanesulfonic acid (HEPES), ethylenediaminetetraacetic acid (EDTA), N-acetylcysteine (NAC), Triton X-100, and calcein acetoxymethyl ester (calcein-AM) were purchased from Shanghai Aladdin Biochemical Technology Co., Ltd. (Shanghai, China). Collagenase D, deoxyribonuclease I (DNase I), diethyl ether, and acetonitrile were obtained from Merck (Darmstadt, Germany). Deferoxamine mesylate, trichloroacetic acid (TCA) and dithiothreitol (DTT) were obtained from Sigma-Aldrich (St. Louis, MO, USA). Fetal bovine serum (FBS), 4,4-difluoro-5-(4-phenyl-1,3-butadienyl)-4-bora-3a,4a-diaza-s-indacene-3-undecanoic acid (C11-BODIPY581/591), MitoSOX Red, and ferritin heavy chain (FTH1) antibody were purchased from Thermo Fisher Scientific (Waltham, MA, USA). 4′,6-diamidino-2-phenylindole (DAPI) was obtained from Solarbio (Beijing, China). Dulbecco’s Modified Eagle Medium/Ham’s F-12 (DMEM/H) high-glucose medium was obtained from HyClone (Logan, UT, USA). Percoll density gradient medium was obtained from Solarbio (Beijing, China); 3-(4,5-dimethylthiazol-2-yl)-2,5-diphenyltetrazolium bromide (MTT), chloroquine, glutathione (GSH), tris(2-carboxyethyl)phosphine (TCEP) hydrochloride, and lithium hydroxide were from Energy Chemical (Shanghai, China). Bicinchoninic acid (BCA) protein assay kit, MitoTracker, and LysoTracker were purchased from Beyotime Biotechnology (Shanghai, China). Hydrogen peroxide solution (3%) was obtained from Guangdong Hengjian Pharmaceutical Co., Ltd. (Jiangmen, China). β-actin antibody was obtained from Cell Signaling Technology (Danvers, MA, USA), and protease inhibitor cocktail was obtained from Amresco (Solon, OH, USA). Calcein was purchased from Shanghai Aomar Biotechnology Co., Ltd. (Shanghai, China).

### 4.2. Isolation and Culture of Primary Mouse Hepatocytes

Male C57BL/6 mice (aged 10–14 w, 16–20 g) were purchased from Hunan SJA Laboratory Animal Co., Ltd. (Changsha, China) and housed in the SPF laboratory of the Experimental Animal Center, Shantou University Medical College with a 12 h light–dark cycle and supplied with a standard pellet diet ad libitum. The study protocol was approved by Shantou University Medical College Animal Ethical Committee (No. SUMC2018-037).

Primary mouse hepatocytes (PMHs) were isolated from C57BL/6 mice using a modified Seglen two-step perfusion method, as we previously described [44]. Briefly, after the mice were sacrificed by CO_2_ inhalation, U-shaped incision along the lower abdomen was performed. Then, the inferior vena cava was separated, and a venous indwelling needle was inserted and fixed. The superior vena cava was also ligated and a small incision was made at the portal vein. EDTA (0.05 mM, flow rate 3 mL/min) for 3 min, protease (3.2 mg/mouse) for 5 min and collagenase D (3.7 U/mouse) for 3–4 min were then infused sequentially at 42 °C. After digestion, the whole liver was removed and placed in the test tube containing collagenase D for further digestion at 42 °C for 5 min. Then, the liver was transferred to a 10 cm dish containing DMEM/H medium (supplemented with 10% FBS) and subjected to mincing to release isolated cells. The cell suspension was filtered through a 200-mesh cell strainer and centrifuged for 3 min at 50× *g* (4 °C). DMEM/H medium (10 mL) was used to re-suspended the cells and then, the cell suspension was slowly added to the 50% Percoll solution. After centrifugation (400× *g* 10 min, 4 °C), the resulting cell pellet was washed twice using PBS and resuspended in DMEM (supplemented with 10% FBS) and cultured for further experiments.

### 4.3. Evaluation of Cell Viability After Treatment with APAP

Freshly isolated PMHs were seeded into 96-well plates at a density of 8 × 10^3^ cells/well. After incubation overnight, the cells were treated with 100 μL culture medium containing different concentrations of APAP (10, 15, 20, 25, and 30 mM) for 2, 4, 6, 8, 10 and 12 h. The cells incubated with equal volume of vehicle served as a control. After exposure, the medium was discarded and replaced with medium supplemented with 1/10 volume MTT (5 mg/mL). Following incubation for 4 h, the medium was aspirated and 100 μL of DMSO was added, gently shaking for 10 min to dissolve the formazan crystals. Absorbance was recorded at 570 nm using a microplate reader (Thermo, USA). The cell viability was determined using the following formula: cell viability (%) = [Absorbance(sample)/Absorbance(control)] × 100%, where Absorbance (sample) represented the reading from the treated cells and Absorbance (control) from the control. Then, a concentration of 20 mM APAP with 12 h exposure was selected for subsequent experiments to induce reproducible cellular injury, consistent with prior literature [34,45] and further validated by the above MTT assay (Figure 1A).

### 4.4. The Effect of DFO, NAC, Chloroquine or Leupeptin on APAP-Induced Hepatotoxicity

To investigate whether ferroptosis, or lysosomal or autophagy activation were involved in APAP-induced hepatotoxicity, iron chelator DFO (0.25, 0.5, 1, 2 mM), NAC (1, 2, 4 mM), autophagy inhibitor chloroquine (25, 50, 75, 100 μM) or protease inhibitor leupeptin (50, 100, 200, 300 μM) were employed in the MTT assay. The concentrations of DFO and NAC were selected based on published hepatoprotective ranges [45]. Similarly, leupeptin and chloroquine doses were chosen to cover documented inhibitory ranges for proteases and autophagy [24,46].

### 4.5. Electron Microscopy

Freshly isolated PMHs were grown on 25 cm^2^ square flasks at 1.2 × 10^6^ cells/mL in 5 mL of culture medium, and exposed to 20 mM APAP, 20 mM APAP + 2 mM NAC, or 20 mM APAP + 2 mM DFO for 12 h. The cells incubated with an equal volume of vehicle served as a control. Then, the cells were scraped and collected after centrifugation at 200× *g* for 3 min. The cell pellets were fixed at 4 °C in 2.5% glutaraldehyde (dissolved in 0.01 M PBS, pH 7.2) for 2 h, and then in 1% oxalic acid for another 1 h in a microwave unit at 37 °C. After dehydration with ethanol, the cells were embedded in LR White. Ultrathin sections (70 nm) were then counterstained with 5% uranyl acetate and lead citrate. Micrographs were obtained with a transmission electron microscope (JEM1400, Tokyo, Japan).

### 4.6. Determination of GSH by High-Performance Liquid Chromatography (HPLC)

Cellular GSH level was determined by using the HPLC system as previously described [47]. Briefly, an aliquot of 200 µL cell lysate was kept at room temperature for 10 min after the addition of 20 µL 0.1 M TCEP. Next, 200 µL PBS (0.2 M, pH 5.0) and 20 µL CMQT (0.1 M) were added and kept at room temperature for 5 min. Then, an aliquot of 110 µL 20% TCA was added, vortexed, and centrifuged at 12,000× *g* for 10 min. The supernatant was then subjected to analysis by HPLC (Agilent 1100, Santa Clara, CA, USA) with an Agilent Zorbax SB-C18 column (4.6 × 150 mm, 5 µm) at 30 °C. The mobile phase was 50 mM TCA (pH 1.65 adjusted with 50 mM lithium hydroxide)-Acetonitrile (85:15, *v*/*v*) at a flow rate of 1 mL/min and the injection volume was 20 µL. The wavelength of the UV detector was set at 350 nm. The standard curve for GSH and the presentative chromatogram are shown in Appendix A.

### 4.7. Determination of Cellular Labile Iron Pool (LIP)

Intracellular LIP was determined by using calcein-acetoxymethyl ester (Calcein-AM). Freshly isolated PMHs were seeded into 96-well plates at a density of 8 × 10^3^ cells/well. After incubation overnight, the cells were treated with either 20 mM APAP, 20 mM APAP + 2 mM NAC, or 20 mM APAP + 2 mM DFO for 12 h. The cells incubated with an equal volume of vehicle served as the control. After exposure, the cells were washed with PBS (0.01 M, pH 7.2), and then 0.1 mL Calcein-AM (0.125 μmol/L in serum-free culture medium) was added and incubated for 15 min. After washing with 0.1 mL PBS, serum-free culture medium (90 μL) was added. Fluorescence intensity (A) was then measured at λex 490 nm and λem 530 nm with a fluorescence microplate reader (SpectraMax Gemini XS, Molecular Devices, Sunnyvale, CA, USA). Finally, 2,2- bipyridine (1 mmol/L, 10 μL) was added to each well and incubated for 60 min. The fluorescence intensity (B) was measured. The content of cellular LIP was determined by the relative increase in fluorescence intensity (B-A).

### 4.8. Lipid Peroxidation Measurements

The oxidation of Cl1 BODIPY served as an indicator of lipid peroxidation. Oxidized Bodipy (Bodipyox) was calculated based on the fluorescence intensity per pixel from the green channel fluorescence images. Freshly isolated PMHs were grown on 35 mm^2^ glass Petri dishes at a density of 3 × 10^5^/dish. The cells were treated with either 20 mM APAP, 20 mM APAP + 2 mM NAC, or 20 mM APAP + 2 mM DFO for 12 h. The cells incubated with an equal volume of vehicle served as a control, and the cells incubated with 50 µg/mL rosup served as a positive control. At the end of exposure, cells were gently washed with PBS, and then incubated with 10 µM C11-BODIPY for 1 h at 37 °C. Following a gentle wash with PBS, the cells were observed under a Zeiss LSM 880 Confocal Laser Scanning Microscope (Carl Zeiss, Oberkochen, Germany).

### 4.9. Measurement of Mitochondrial Superoxide with MitoSOX-

The production of superoxide in mitochondria was visualized with MitoSOX Red. Freshly isolated PMHs were grown on 35 mm^2^ glass Petri dishes at a density of 3 × 10^5^/dish. The cells were treated with either 20 mM APAP, 20 mM APAP + 2 mM NAC, or 20 mM APAP + 2 mM DFO for 12 h. The cells incubated with an equal volume of vehicle served as a control, and the cells incubated with 50 µg/mL rosup served as a positive control. After exposure, the cells were washed with PBS, and then incubated with serum-free culture medium containing 100nM Mito Tracker Green for 30 min at 37 °C. Following a gentle wash with PBS, culture medium containing 5 µM MitoSOX Red was added to each culture, and incubated for 10 min. After washing with PBS, the cells were observed under a Zeiss LSM 880 Confocal Laser Scanning Microscope (Carl Zeiss, Oberkochen, Germany).

### 4.10. Western Blotting Analysis of Ferritin

Freshly isolated PMHs were seeded into 96 well plates at a density of 8 × 10^3^ cells/well. After treatment with either 20 mM APAP, 20 mM APAP + 2 mM NAC, 20 mM APAP + 2 mM DFO, 20 mM APAP + 100 μM CRO, or 20 mM APAP + 200 μM Leupeptin, the cells were extracted in lysis buffer [Tris-HCl (20 mM, pH 7.2), EDTA (1 mM), and cocktail (104 mM PMSF, 0.1 mM E-64, 0.08 mM Aprotinin, and 0.1 mM Leupeptin)] for 30 min on ice, and insoluble material was removed by centrifugation at 12,000× *g* for 30 min. After determination of protein contents with the BCA Protein Assay Reagent Kit, the cell lysate (30 μg) was separated on 12% sodium dodecyl sulfate-polyacrylamide gel electrophoresis (SDS-PAGE) gels and transferred to polyvinylidene fluoride (PVDF) membranes. After blocking with 5% non-fat milk, the membranes were incubated with specific anti-ferritin heavy chain 1 (FTH1) antibody (1:2000 dilution), or β-actin antibodies (1:250 dilution) at 4 °C overnight. Next, the membranes were washed with TBST (150 mM NaCl, 20 mM Tris, 0.1% Tween 20, pH 7.2) followed by incubation with goat anti-rabbit IgG secondary antibodies linked to horseradish peroxidase at a 1:4000 dilution, at room temperature for 2 h. After washing with TBS (150 mM NaCl, 20 mM Tris, pH 7.2), immunoreactive bands were visualized by means of enhanced chemiluminescence and detected by ChemiDoc XRS Image System (Bio-Rad Laboratories, Hercules, CA, USA). Densitometric analysis of the scanned bands was performed by using Image J software (Version: V1.8.0.112). The results were expressed as the percentage of corresponding negative control. All data are representative of at least 4 independent experiments.

### 4.11. Determination of Iron Released from Ferritin Induced by Oxidative Injury Using Calcein Fluorescence

Ferritin (0.25 mg/mL) was treated with hydrogen peroxide (H_2_O_2_, 10 μM), to which calcein (0.5 μM) was added to probe free iron. The fluorescence intensities were then measured at λex 490 nm and λem 530 nm with a fluorescence microplate reader, at different time points (15, 30, 60, 120, and 240 s). Iron binding to calcein quenches its fluorescence [48]. Therefore, the decrease in fluorescence intensity indicated the increased level of iron in the solution. Solution without hydrogen peroxide was used as a control.

### 4.12. Statistical Analysis

Data are presented as the means ± standard deviation (SD). Statistical analysis was performed by using GraphPad Prism software (version 10.1.1 for Windows). The normality of data was evaluated by the Kolmogorov–Smirnov test. Analysis of Variance (ANOVA) with post hoc Tukey’s multiple comparisons test was used for comparison of means between groups. Each group included 3–6 biological replicates per experiment, ensuring 90% power to detect a significant difference (two-sided test, α = 0.05).

## 5. Conclusions

In summary, this work demonstrates that ferroptosis contributes to APAP hepatotoxicity via GSH depletion and iron overload characterized by ferritin oxidative destruction. Furthermore, it also elucidates a vicious cycle in APAP hepatotoxicity where initial oxidative stress causes ferritin degradation, iron release, and further oxidative damage. The findings support exploring iron chelation as adjunct therapy and highlight ferritin as a potential therapeutic target for preventing APAP hepatotoxicity.

## Figures and Tables

**Figure 1 ijms-26-07585-f001:**
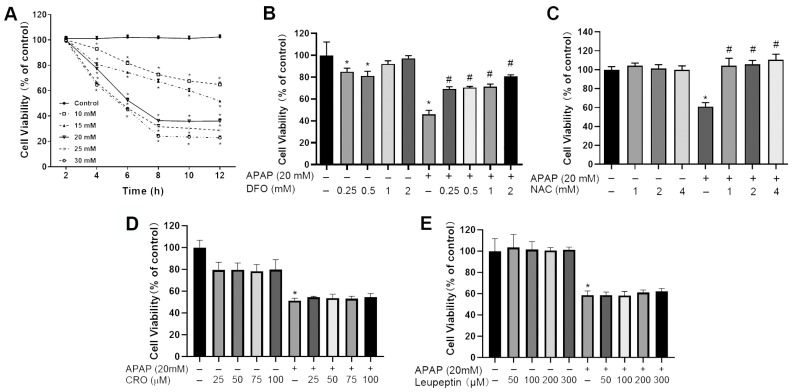
The effect of different concentrations of APAP on the cell viability of PMHs for various durations of time (n = 6). (**A**) Incubation of PMHs with 10–30 mM APAP and (**B**) DFO (0.25–2 mM) could alleviate the toxic effect of 20mM APAP in a concentration-dependent manner (*p* < 0.05); (**C**) NAC (1–4 mM) could alleviate the toxic effect of 20mM APAP in a concentration-dependent manner (*p* < 0.05); (**D**,**E**) chloroquine (CRO, 25–100 μM) or leupeptin (50–300 μM) treatment did not significantly inhibit the toxic effect of 20 mM APAP (*p* > 0.05). * *p* < 0.05, compared with the corresponding control group; # *p* < 0.05, compared with APAP treatment group.

**Figure 2 ijms-26-07585-f002:**
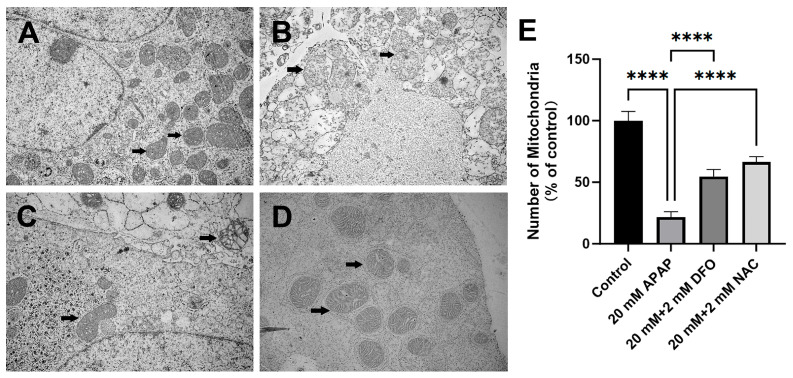
APAP-induced PMH damage shows ultrastructural features of mitochondria morphological changes. (**A**), Representative electron micrograph of control cell showing intact mitochondria (black arrow); (**B**), APAP-treated PMHs showing spherical mitochondria with cristae rupture or large vacuoles (black arrow); DFO-treated cells (**C**) and NAC-treated cells (**D**) showing restoration of cell nuclear membrane integrity and mitochondrial structures (black arrow) (×12,000); (**E**) mitochondrial counts from 4 replicates (5 distinct fields of view/replicate), **** *p* < 0.0001 vs. APAP-treated group.

**Figure 3 ijms-26-07585-f003:**
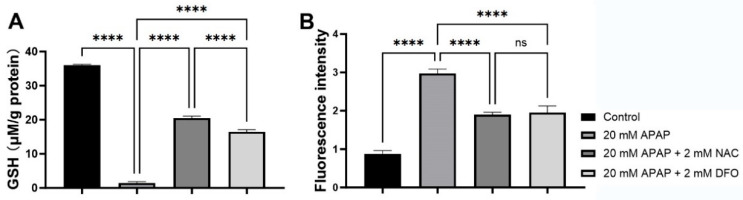
Changes in GSH and LIP contents in PMHs after treatment with 20 mM APAP (n = 3). (**A**), NAC (2 mM) or DFO (2 mM) could significantly inhibit the decrease in GSH level induced by 20 mM APAP (*p* < 0.0001). (**B**), NAC (2 mM) or DFO (2 mM) could significantly inhibit the increase in LIP level (indicated by fluorescence intensity) induced by 20 mM APAP (*p* < 0.0001). Between-group difference: **** *p* < 0.0001.

**Figure 4 ijms-26-07585-f004:**
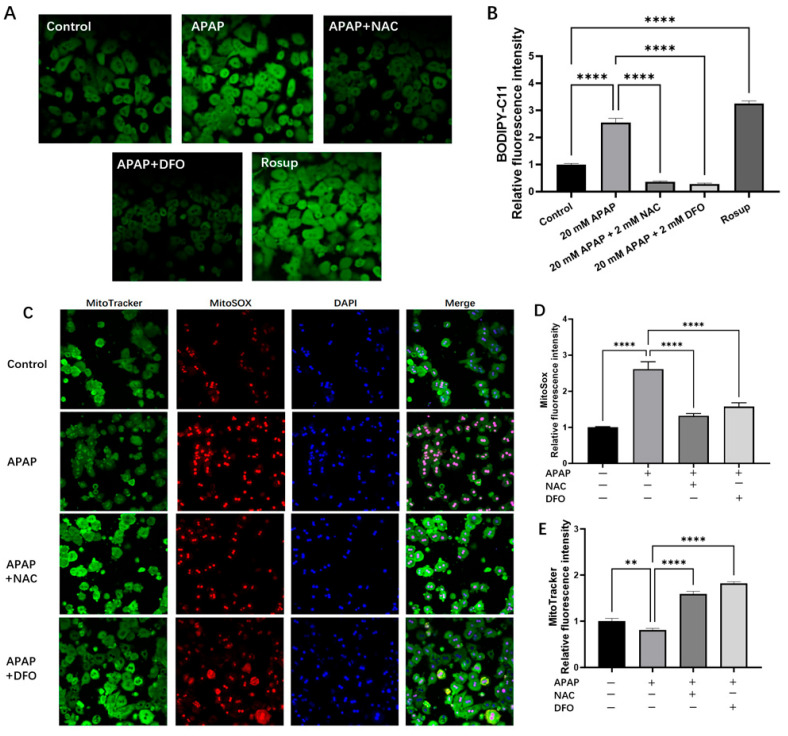
The effects of NAC and DFO on the increased intracellular lipid peroxidation and mitochondrial superoxide production induced by APAP (×200). (**A**) Representative images of lipid peroxidation level of PMHs stained by BODIPY-C11 after various treatments. Rosup treatment group served as positive control in this experiment. (**B**) The relative fluorescence graph of BODIPY-C11 (mean ± SD, n = 3); (**C**), mitochondrial mass was measured by MitoTracker green; treatment with either 2 mM NAC (**D**) or 2 mM DFO (**E**) attenuated the red fluorescence intensity while restoring green fluorescence intensity. ** *p* < 0.01 and **** *p* < 0.0001.

**Figure 5 ijms-26-07585-f005:**
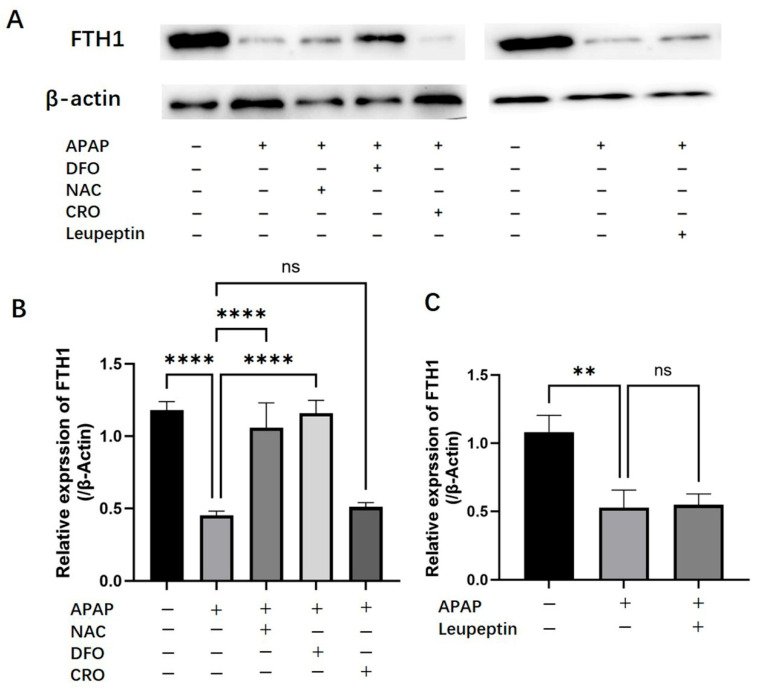
Effects of DFO, NAC, CRO, or Leupeptin on APAP-induced ferritin-depleted PMHs. (**A**), Representative Western blot images of FTH1 levels. (**B**) Quantitative analysis of FTH1 protein levels in control, APAP-treated (20 mM), APAP + NAC (2 mM), APAP + DFO (2 mM), and APAP + CRO (100 μM) groups. (**C**) Quantitative analysis of FTH1 protein levels in control, APAP-treated (20 mM), and APAP + Leupeptin (200 μM) groups. Data are presented as mean ± SD. n = 3, ** *p* < 0.01 and **** *p* < 0.0001.

**Figure 6 ijms-26-07585-f006:**
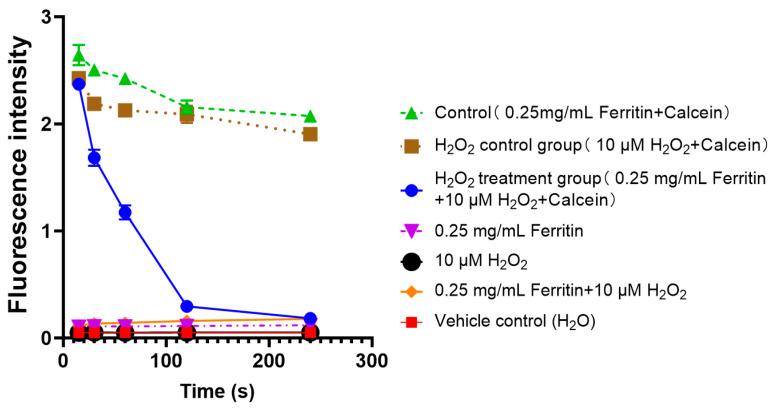
Iron release from ferritin under oxidative stress (n = 3).

## Data Availability

The raw data supporting the conclusions of this article will be made available by the authors on request.

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
