# Peer review of "Oxidative Ferritin Destruction: A Key Mechanism of Iron Overload in Acetaminophen-Induced Hepatocyte Ferroptosis"

_ijms, 2025, doi:10.3390/ijms26157585_

Round 1
Reviewer 1 Report
Comments and Suggestions for Authors
The manuscript ijms-3777503 with the title “Oxidative Ferritin Destruction: A Key Mechanism of Iron Overload in Acetaminophen-Induced Hepatocyte Ferroptosis” investigates a topic of great interest, especially in the post-COVID-19 era, specifically, how oxidative stress induces ferritin destruction and how this process may serve as an important link between primary oxidative injury and secondary iron overload in APAP-induced hepatotoxicity.
The authors approached several methods on an in vitro model to highlight these aspects, obtaining statistically significant results. The study also presents a novel element, in particular iron overload during APAP toxicity. Furthermore, the hypothesis was supported by the protein expression of FTH1.
The manuscript is well-structured, admirably organized, and below are several comments for improving this manuscript:
Major comments:
-The purpose of the study should be more clearly highlighted at the end of the Introduction section.
-Figure 6: Only 5 groups can hardly be distinguished, but 7 conditions appear in the legend. Maybe it would be more suggestive a graphical representation with a larger width and/or with the conditions coloured differently to be better distinct compared to the overlay data or standard deviation which is already in black. Also, the figure legend should be expanded so that it can be understood by any reader without having to read all the text in the results.
-Discussions section: How was the concentration range for CRO, NAC, DFO and leupeptin chosen? But also, the working concentrations presented for APAP [20 mM]?
-Discussion section: How was the exposure time chosen for the compounds used in the present study?
-Lines 369-379: For the determination of GSH how was the quantification performed on HPLC? If a standard curve was used, this should be added in the supplementary materials and including a chromatogram example for the sample(s).
-Statistical analysis (lines 442 - 446): Whether data from technical repetitions were averaged?
-Statistical analysis (lines 442 - 446): What was the power of used statistical tests?
Minor comments:
-Line 226: I did not find the supplementary materials Figure 1.
-Lines 229-232: "This attenuation may reflect saturation of cytochrome P450-mediated metabolic activation, as the conversion of APAP to its toxic metabolite NAPQI is ultimately limited by the enzymatic capacity of hepatic CYP isoforms, particularly CYP2E1. This part of the discussion would be appropriate to be correlated with existing data from the scientific literature on how CYP2E1 activity is influenced under similar toxicity conditions.
-Discussions section: The discussions are concise and clearly presented. I have only one suggestion, especially since the authors already own the infrastructure, in future studies the results can be elegantly correlated with the level of protein expression but also enzyme activity for CYP2E1 or other cytochrome P450 family proteins. Let's not forget that the ultimate effector in the central dogma and all the living world is the protein and its specific activity.
-The methodology part would be better organized in subsections.
Reviewer 2 Report
Comments and Suggestions for Authors
The research work is relevant considering the ongoing battle against drug toxicity especially Acetaminophen. The authors provided adequate research rationale for the study with sufficient methodologies. The results was well articulated and presented using graphics. This gave the scientific merit for the research work. However, the manuscript still require minor revision to further improve its quality. These include;
- The authors should include brief methodology in the abstract session.
- correct the typo for TAcetaminophen spelling . line 34.
- correct the technical errors of the values presented in Line 98-99.
- Figure 6 graph is incomplete. The value for 0.25 mg/ml Ferritin and 10 uM H2O2 are not presented in the graph.
- The materials and methods section needs to be subtitle for better clarity. The authors should subtitle each of the paragraphy in the material and method section.
- The references need to be check for uniformity and punctuation errors.
- The figures should be formatted in line with the authors instruction guide.
The manuscript still requires language editing.
Reviewer 3 Report
Comments and Suggestions for Authors
Oxidative Ferritin Destruction: A Key Mechanism of Iron Over load in Acetaminophen-Induced Hepatocyte Ferroptosis
Title:
The title contains the keywords that introduce the experimental work through the sentences Oxidative Ferritin Destruction, Iron Overload, Acetaminophen-Induced, and Hepatocyte Ferroptosis, so there are no comments.
Abstract:
This section should be reconstructed because it consists of a brief introduction, methodology, results, conclusion, and objective.
- On line 22, add the word "in vitro" in italics.
Introduction:
Contains the necessary information to support the title, subjects and methods, results, and discussion. However, observations and suggestions regarding form and content are listed below:
- On line 35, correct the word TAcetaminophen.
- On line 45 mentions N-acetyl cysteine, this compound and its mechanism of action can be described in more detail to prevent hepatotoxicity caused by an overdose of acetaminophen.
Materials and Methods:
The methodology, equipment, and conditions used in this work are adequate and explained in detail, allowing for repeatability by other researchers worldwide. The mouse protocol was evaluated by an ethics committee. Observations are listed as follows:
- On line 328, correct CO2inhalation with a space.
- On line 344, change the word "various" to "different."
- On line 359, correct cm2 with a superscript.
- On line 393, correct the word measurements-The to "measurements. The"
- On line 396, correct 35mm2 with a superscript.
- On line 435, correct H2O2 with a subscript.
- On line 439, correct fluorescence(Soe-Lin), leaving a space.
Results:
The results are reported in a clear and understandable manner, demonstrating the importance of each study conducted. These results may be necessary for other researchers worldwide. The figures are clear and easy to interpret. Figure 2 clearly and qualitatively demonstrates the induced damage and morphological changes in mitochondria. Figure 4 also shows the effects of N-acetylesteine and deferoxamine on the increase in intracellular lipid peroxidation and induced mitochondrial superoxide production. However, the following observations on form and substance are listed:
- Figures 1B, 1C, 1E, and 1F are not explained.
- Continue the letters in Figure 1 (A, B, C, D, E).
- I recommend that in Figures 1, 2, 3, 4, and 5, only the name of the figures be included, and the results described therein be placed in the results section and not in the figure caption.
- On lines 128, 129, 130, and 131, the information found in the caption of Figure 2 is repeated.
- On lines 167 and 171, remove a space (check throughout the document).
- On line 183, separate D,E,
- On line 187, the Western blot technique is mentioned; however, it is not described in the methodology.
- On line 206, change ml to mL
- On line 207, change the word seconds to s
Discussions:
The discussion of the results is clear and pertinent, so only formal observations are listed.
- On line 227, correct the word time-
- On line 258, separate Figure 2C,D)
- On line 270, separate Fe³⁺ (Li et al., 2022)
- On line 283, italicize the word In vitro
- Lines 293 to 298 are part of the conclusions
Conclusions:
The document does not present conclusions.
References:
The references are adequate and relevant to this document; however, I believe some of them should be updated.
Reviewer 4 Report
Comments and Suggestions for Authors
Major comments
This manuscript presents in vitro results on the toxicity of paracetamol. The work is clear and well presented. The major shortcoming is the lack of data on the expression or activity of P450 2E1, which is the key enzyme in the formation of the toxic metabolite.
1- This study lacks a section on the expression/activity of CYP2E1, which is essential for the bioactivation of APAP. This can be done by Western blot or even by measuring the specific microsomal activity of this cytochrome. Alternatively, although this is undoubtedly more complex, a NAPQI assay could be performed. At the very least, a bibliographic analysis showing that the culture conditions of primary hepatocytes allow the expression and activity of this cytochrome
2- A semi-quantitative analysis of the images generated by transmission electron microscopy would allow for better quantification of the effects
3- The introduction or discussion lacks a reminder of the metabolic pathways of APAP, other than its conversion to NAPQI. In particular, mention its glucuronoconjugation and biotransformations by P450 leading to other oxidized derivatives than NAPQI.
4- The number of independent replicates must be specified for the different tests.
Detailed comments
L34: “Acetaminophen” and not “TAcetaminophen”
L37-: a mention on the formation of NAPQI, which is decisive in its toxicity, is missing here
L44: also mention metabolic induction due to ethanol
L51: see comment 1.
L97 and the entire manuscript: remove insignificant figures to improve the readability of the manuscript; for example, replace “101.07±1.45%” with “101.1±1.5%.”
L117 and the entire manuscript: add the number of replicates in the figure captions.
L126-135: see comment 3
L212-223: unnecessary repetition of the introduction
L229-232: see comment 1
L300: the subheadings of the paragraphs in this section are incorrectly displayed: “4.1- Reagents”
L442-446: see comment 4
Round 2
Reviewer 1 Report
Comments and Suggestions for Authors
I really appreciated all the authors' responses (especially the statistical comments), which were clear and improved the quality of the manuscript. However, I have a few minor remarks:
- the logic by which the working concentrations in the present study were selected should at least be briefly mentioned in the discussion or methods.
- also, the same for the exposure time.
- there is no reference in the text for Figure S3, this should be included, maybe in the methodology for GSH.
Reviewer 4 Report
Comments and Suggestions for Authors
Thank you for the clarifications provided in your manuscript. I understand that at this stage of the work it is difficult to characterize the expression of P4502E1 in the system used, and I accept your references as sufficient evidence of this capacity. I have no further comments to make.